# Does Increasing Natural Gas Demand in the Power Sector Pose a Threat of Congestion to the German Gas Grid? A Model-Coupling Approach

**Philipp Hauser [1],\* , Sina Heidari [2] , Christoph Weber [2] and Dominik Möst [1]**

[1] Chair of Energy Economics, Faculty of Economics and Business Management, Technische Universität Dresden, D-01062 Dresden, Germany; dominik.moest@tu-dresden.de

[2] Chair for Management Science and Energy Economics, University of Duisburg-Essen, D-45127 Essen, Germany; sina.heidari@uni-due.de (S.H.); christoph.weber@uni-due.de (C.W.)

\* Correspondence: philipp.hauser@tu-dresden.de

**Abstract:** This study aims to investigate the possible congestion in the German natural gas system, which may arise due to an increase in the gas consumption in the power sector in extreme weather events. For this purpose, we develop a two-stage approach to couple an electricity model and a natural gas network model. In this approach, we model the electricity system in the first stage to determine the gas demand in the power sector. We then use the calculated gas demand to model gas networks in the second stage, where we deploy a newly developed gas network model. As a case study, we primarily evaluate our methodological approach by re-simulating the cold weather event in 2012, which is seen as an extreme situation for the gas grids, challenging the security of supply. Accordingly, we use our coupled model to investigate potential congestion in the natural gas networks for the year 2030, using a scenario of a sustainable energy transition, where an increase in the gas consumption in the power industry is likely. Results for 2030 show a 51% increase in yearly gas demand in the power industry compared to 2012. Further, the simulation results show a gas supply interruption in two nodes in 2012. In 2030, the same nodes may face an (partial) interruption of gas supply in cold winter days such as the 6th of February 2012. In this day, the load shedding in the natural gas networks can increase up to 19 GWh$_{th}$ in 2030. We also argue that the interrupted electricity production, due to local gas interruptions, can easily be compensated by other power plants. However, these local gas interruptions may endanger the local heat production.

**Keywords:** coupling of energy sectors; gas networks; electricity and heat markets; energy security

## 1. Introduction

With increasing feed-in from intermittent renewable production from wind and solar energy resources, increasing demand for flexibility in the power system is anticipated. One way to provide the desired flexibility on the generation side is an enhanced utilization of natural gas-fired power plants. Additional gas capacities would seem to be the most likely option as coal capacities are to set to be phased out to meet decarbonization targets. Consequently, an increase in the demand for natural gas in the power industry is likely. This increase would strengthen interdependencies between the electricity and natural gas sectors. In 2014, the power sector already accounted for 27% of the total natural gas consumption in EU28 and 22% in Germany [1]. An enhanced interdependency would suggest that weather-related uncertainties from the electricity system, primarily wind and solar feed-in, can profoundly affect the natural gas system in the short run. More specifically, we explore these potential impacts using the example of a *Dunkelflaute* [2]. A Dunkelflaute, occurring with the highest probability

in winter months, refers to a long period of low feed-in from intermittent renewables. The incidence of such an event puts a strain on the usage of other flexibility options in the electricity system such as pump storage or demand-side flexibilities. Moreover, low temperatures during a cold Dunkelflaute cause an increase in heat demand. Figure 1 shows the daily average residual load (residual load = actual load – feed-in of fluctuating renewable energies) of electricity as well as the daily average temperature in Germany in 2012. During the three day period from the 6th to the 8 February, record low temperatures were recorded with the lowest temperature being reached on the 6th of February. On this day, the daily average residual load peaked at 73 GWhel while temperatures fell to minus 13 degrees Celsius. During this historical event, gas supply shortfalls affected many gas power plants in Germany with an aggregated capacity of 3.5 GW. As several studies discuss the economic, technical and political reasons for these shortfalls, e.g., [3,4], we take this as an occasion to study the resiliency and energy security of the future gas and electricity system under similar weather conditions as those experienced in 2012.

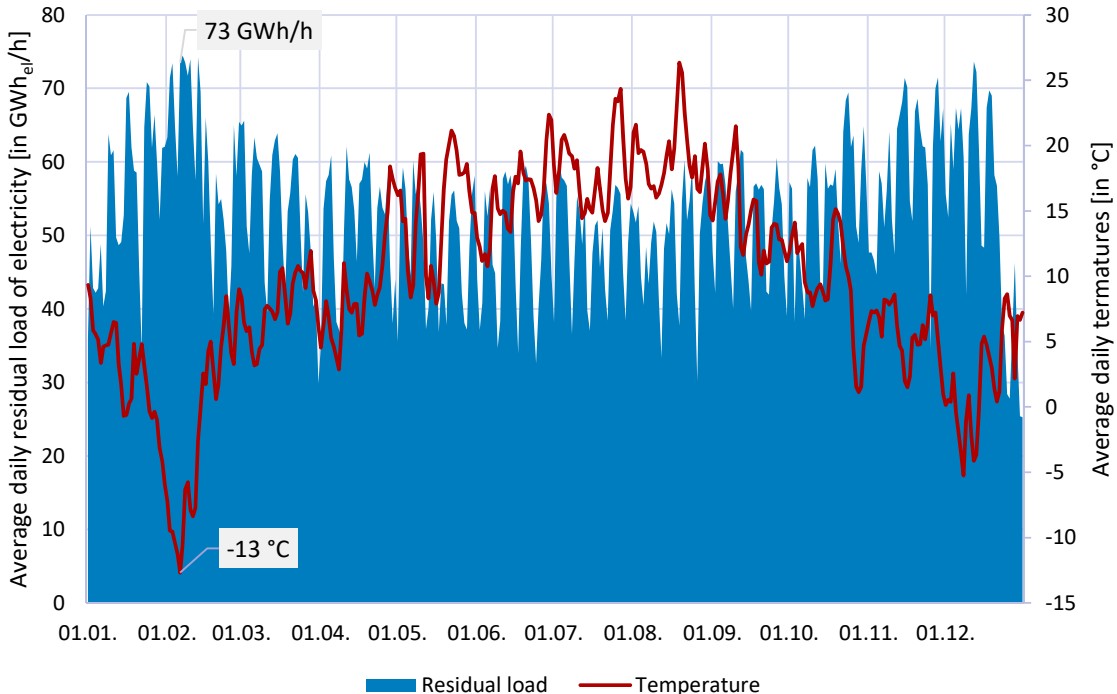

**Figure 1.** Temperature and electrical residual load profiles in 2012. Source: authors' illustration based on [5–7].

The security of energy supply can be investigated from a broad range of prespectives [8,9]. Among them, we can find (geo-)political analyses, which focus on qualitative insights as well as techno-economic analyses, which focus on quantitative measures to evaluate security of supply (SoS). Another dimension for analysing SoS distinguishes between comodities, i.e. natural gas, oil or electricity. In this context, electricity constitutes a particular case, as it is not storable. Yet also for gas, large-scale storage is typically restricted to a limited number of large gas storage facilities in the network. A further major challenge consists in identifying appropriate indicators for evaluating SoS. [10]. For natural gas, the SoS literature mainly deals with geopolitical issues such as gas disputes, etc. [11–15]. By contrast, this paper sheds light on the local security of gas supply in the German gas grid.

With increasing demand during a cold Dunkelflaute, one may expect an increased deployment of gas-fired plants in the power and heating sector to cover demand, creating an increased demand for natural gas. Natural gas demand in the power industry is set to gain more importance with the changing structure of the future German energy systems stemming from a nuclear phase-out by 2022

as well as an accelerated coal phase-out to be carried out during the 2020s and 2030s. The further operation of gas-fired power plants with high demand peaks can lead to incidences of congestion in the natural gas network and could potentially endanger the SoS. Adding to this concern, several studies suggest that climate change may result in severer winter storms in Europe [16], which could raise energy demand and consequently increase the incidence of a Dunkelflaute. Therefore, investigating electricity-triggered congestion in natural gas networks is becoming more relevant. Assessing the interdependencies between the electricity, heat and gas sectors is complex and the academic literature provides a broad range of methods and approaches, whose application depends on the respective research question.

Model-based approaches are used in a growing share of academic literature, largely to study energy security-related questions. Among them, a particular focus lies on the interaction of coupled gas and electricity sectors. In general, we identified three common approaches: integrated models, stylized models and model-coupling approaches. The first group, integrated models, considers energy systems within a single equilibrium model [17–19]. These models are used to evaluate sectoral interdependencies endogenously, e.g., technology diffusion in heating applications and its effects on primary energy demand. However, due to the rudimentary representation of energy sectors, these models usually lack a detailed spatial (sub-national) and temporal resolution based on annual figures at a country level neglecting energy grid infrastructure. Therefore, an assessment as to whether reliable energy transport (electricity or gas) is ensured is not discussed in detail. The second group of studies proposes stylized models that focus on interdependencies between different energy sectors, e.g., the power and gas sectors. These models reduce complexity by limiting the considered energy sectors, which allows for considering further technical constraints [20–25], considering strategic behavior [26] or dealing with uncertainties by using various methods like stochastic programming. An example of using integrated models to study the interdependencies between the gas and electricity sector can be found in [21], where the authors investigate disruption scenarios based on an exemplary network topology. Uncertainties are addressed using a robust day-ahead scheduling model for the optimal coordinated operation of an integrated gas and electricity system in [27]. However, as these models focus on the nexus of a coupled gas and power sector, they are mainly applied to case studies using stylized topologies with a limited number of nodes and branches. Finally, a common approach comprises partial energy system models, e.g., power market models that cover additional sectors such as the gas sector in two different ways. One option is to use scenario-based parameters. This approach is used by the authors in [28] as they develop a security-constrained unit commitment model to analyze the short-term impact of natural gas prices on power generation scheduling in the USA. In order to analyze the impact of gas supply shortfalls, they vary the gas price in four scenarios up to an increase of 15% and show that gas power plants would lose their competitive advantage. In contrast, the authors in [29] investigate the impact of an electrification scenario on the gas sector. A second option in deploying partial energy system models involves model coupling, sometimes also called sequential modeling, where two or more models iterate input and output parameters in order to cover all sector-specific aspects at a highly detailed level [30,31]. We follow the latter approach in the present study.

None of the literature reviewed investigates German electricity and gas sectors with a significant level of detail regarding international and national data. In this study, we simulate the European electricity market to determine the electricity exchange for the simulation of the domestic power market. Unlike most of the previous studies, we consider local heating markets and their corresponding regional heating networks to offer a more precise representation of combined heat and power (CHP) plants with a high spatial resolution. We consider this aspect as important, as the heating sector is essential and a large share of produced heat is natural gas-based (natural gas accounted for 45% of total German heat production in 2015 [32]). For the natural gas sector, we provide a high temporal, sectoral and regional resolution regarding demand, production and infrastructure, enabling us to study regional threats to the SoS. Our results indicate that the gas network ensures energy security for

the German Energiewende as it provides a resilient energy transport infrastructure even in extreme weather situations to supply natural gas to gas power plants.

The outline of the paper is as follows: Section 2 presents our methodology by giving a general overview of our two-stage model coupling approach. Additionally, we briefly describe the major features of the models deployed, JMM and GAMAMOD-DE. The section ends with an overview of the study design and data collection. Section 3 presents the results of both the backtesting for the year 2012 and simulation for the year 2030. In Section 4, we discuss the limitations of the study before we conclude in Section 5.

## 2. Methodology

### 2.1. General Approach and Interface between the Electricity and Gas Models

We present an overview of the methodology in Figure 2. The present approach is based on two stages. In the first stage, we model electricity markets using the "Joint Market Model" (JMM) to determine the commitment schedules as well as the fuel consumption of gas-fired power plants. In the second step, we simulate natural gas flows within the German natural gas networks using the "Gas Market Model Germany" (GAMAMOD-DE) considering the demand for natural gas in the power industry, which is determined in the first stage, to investigate possible incidences of congestion. The following section provides a detail description of each stage.

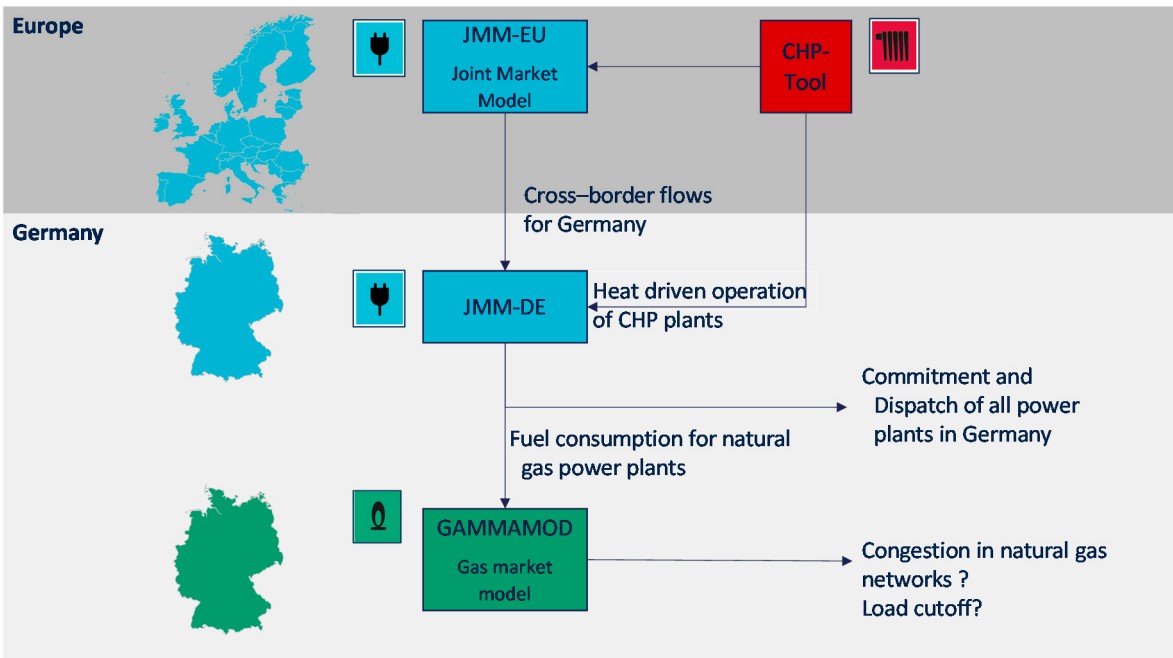

**Figure 2.** Overview of coupling JMM and GAMAMOD. Source: authors' illustration.

### 2.2. The Electricity Model—Joint Market Model (JMM)

To model the electricity market, we use the JMM, initially developed within the EU-projects WILMAR and Decision Support for Large Scale Integration of Wind Power (SUPWIND) and used in various studies such as [33–38]. JMM is an optimization-based market model for unit commitment and dispatch based on the hourly representation of generation, demand, and transmission, combining technical and economic aspects. It includes a day-ahead market for physical delivery of electricity, an intra-day market for handling deviations between expected and realized production as well as a day-ahead market for automatically activated reserve power and an intra-day market for positive secondary reserve power with exogenous demands. The model is formulated as a deterministic linear or mixed-integer program.



To tackle the computational burden as well as to maintain a convex optimization problem, we use the deterministic linear version of JMM to model the European electricity markets. In this case, we cluster the generating units based on their technical properties. As a result of the simulation, we calculate the electricity exchange (import and export) for the specific market region of Germany and Luxemburg. Once the electricity trade for this market region is determined, we use the deterministic mixed-integer linear version of JMM to compute the commitment schedules for each German power plant (cf. Figure 2). Although using a mixed-integer approach proves suitable for calculating the commitment schedules, the output prices correspond to the operating cost of the most expensive cleared unit in the market clearing process. Consequently, we use the linear formulation of the model to determine the electricity prices within the considered region, which provides for an enhanced estimation of German electricity prices.

In the present study, the JMM uses an exogenous heat extraction time series generated with the CHP model presented in [12,13]. In this approach, CHP units are connected to district heating networks. Each of the heating networks acts as a local heating market, in which the attached CHP units form a merit order curve. With a given temperature-dependent demand, the heating markets are cleared, allowing for the calculation of thermal operation of CHP units. Searching for detailed data for all the heating networks inside Germany and neighboring countries is challenging due to the lack of information and a large number of existing district heating networks. In the present work, we limit our study to 28 heating networks inside Germany and a few selected networks in neighboring countries. The rest of the CHP units without any specific heating network are assumed to be connected to a few aggregated heating networks, to which the remaining heat demand of each country is allocated. We consider these aggregated heating networks to allow for a differentiation based on fuel types.

Once the hourly commitment and dispatch and consequently fuel usage of gas-fired units are determined, we aggregate these hourly time series on a daily basis to provide a suitable demand time series for the natural gas market model. Using the individual gas power plant efficiency, the electric gas power plant output is converted into thermal natural gas demand. These data are used as part of the input gas demand in the natural gas market model.

### 2.3. The Gas Model—Gas Market Model for Germany (GAMAMOD-DE)

The gas market model for the German gas grid (GAMAOD-DE) is a linear optimization model that minimizes total system costs, i.e. costs for domestic production or import of natural gas, transport of natural gas within Germany as well as costs for load shedding. The model is run with a daily resolution for one year. Figure 3 illustrates the pertinent input and output data.

The model covers the German gas transport network, encompassing more than 1400 pipelines and 900 nodes. Connections to the European gas network are covered by explicitly modeling cross-border pipelines. The model considers technical characteristics of pipelines, i.e. volume, gas quality and pressure, by converting physical gas flows to energy flows and by following a transport system approach. The model includes gas storages with specific characteristics of injection and withdrawal rates for three different kinds of gas storages: aquifer, salt caverns and depleted gas fields. We model exports exogenously based on historical export volumes according to [39] and imports endogenously. Cross-border import prices are based on monthly historical time series taken from [40].

The gas demand is represented with a high temporal and spatial resolution comprising the consumption in private households, industries and the power sector. Natural gas demand for households is calculated based on a bottom-up approach using daily average temperatures for more than 400 regions in Germany according to the NUTS3 level. The industrial gas demand is based on [41] and differentiated among several industries, e.g., paper, aluminum, steel, chlorine, ceramics and glass, tobacco and food. Following a top-down approach, industrial gas demand is allocated to NUTS3-regions by taking German industry centers and locations into account. For the study, natural gas consumption in the power sector is based on the consumption of gas-fired power plants calculated by the JMM model as shown in Figure 2.

A detailed description of the model formulation can be found in [42]. The database of GAMAMOD-DE is described in [32].

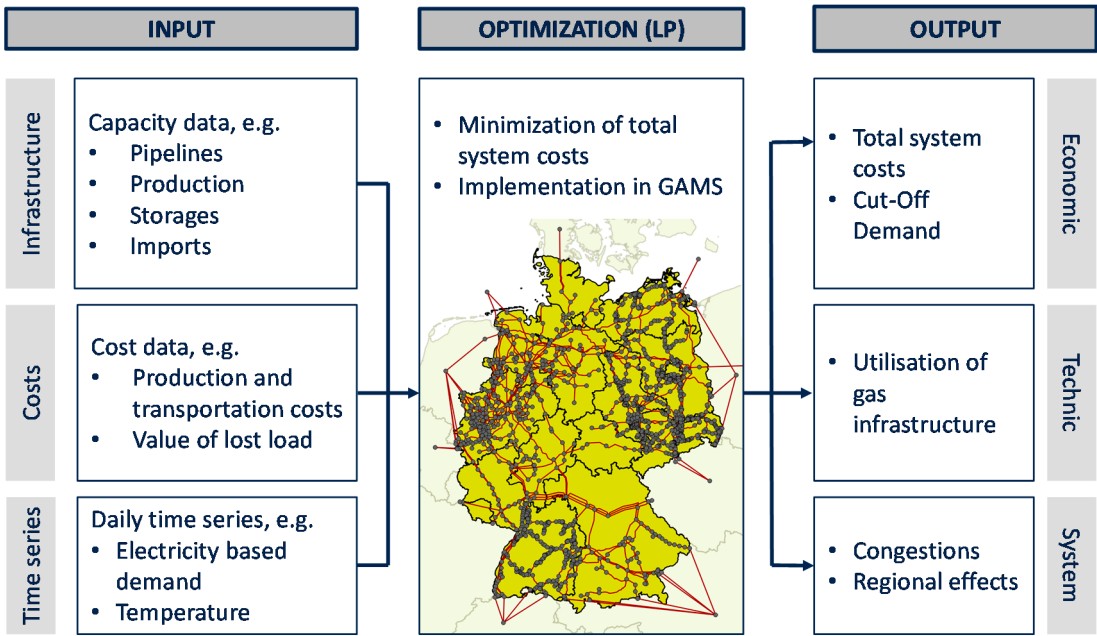

**Figure 3.** Structure of the model GAMAMOD-DE. Source: authors' illustration based on [30].

## 2.4. Benefits and Drawbacks of the Selected Approach

Compared to other approaches that study energy security-related questions, the model coupling approach has significant advantages. One great benefit of the model coupling approach is to enable the individual models to use data with high spatial (sub-national) and temporal resolution for both the natural gas and electricity systems. Furthermore, we can consider specific technical constraints in the formulation, e.g., ramping of generation units, while maintaining a convex problem. On the other hand, as we discuss in Section 4, the model coupling approach requires an iterative process. Considering the main topic of the current paper, we believe that our approach is the best way to cope with the problem of investigating the local SoS. However, undergoing an iterative process using models with high resolution data is extremely time-consuming. Furthermore, it is not ensured that the iteration process converges to one optimal solution, without jumping between different equilibria (cf. [30]). As discussed earlier in Section 1, as an alternative approach, integrated models can be used to tackle the problem of time expenditure. Still, using high resolution data in the integrated models are challenging. As with many detailed models, data availability and quality is a critical issue. The database used in this study has been compiled from publicly available sources (cf. [32]). Thereby data sources have been carefully selected and quality checks have been carried out, yet some inaccuracies may persist, especially since power plant efficiencies and transport capacities in the gas network had to be approximated using engineering estimates.

## 2.5. Research Design and Data Assumptions

The overall objective of this study aims to investigate possible congestion in the natural gas network in 2030. We initially evaluate our methodological approach by re-simulating the power and natural gas markets (backtesting) for a historical year. We select the year 2012, as its cold winter can be seen as a case study for extreme weather events for the electricity and gas sector where disruptions in the natural gas supply occurred.

This study focuses on the German electricity and gas sectors by using a two-stage model coupling approach. To investigate the German electricity system the power exchanges between Germany and its

neighboring countries are intially determined. For the electricity system, historical data regarding the power exchanges are obtained from ENTSO-E [6]. For the purpose of backtesting for 2012, we use these data as an exogenous input for imports and exports to Germany. However, with the current ongoing changes in the electricity system structure, it is improbable that the time series of electricity exchange will remain similar to historical ones. Therefore, we initially simulate the European electricity markets to determine the time series of electricity exchange for Germany. Using these calculated time series, we subsequently simulate the German electricity market in 2030.

In order to simulate the future electricity sector, we construct a scenario for the year 2030. The scenario is based on the assumptions in the TYNDP 2018 [43] for European countries and scenario C of the German Grid development plan (NEP 2030 [44]). The aggregated power generated capacities for Germany are displayed in Figure 4.

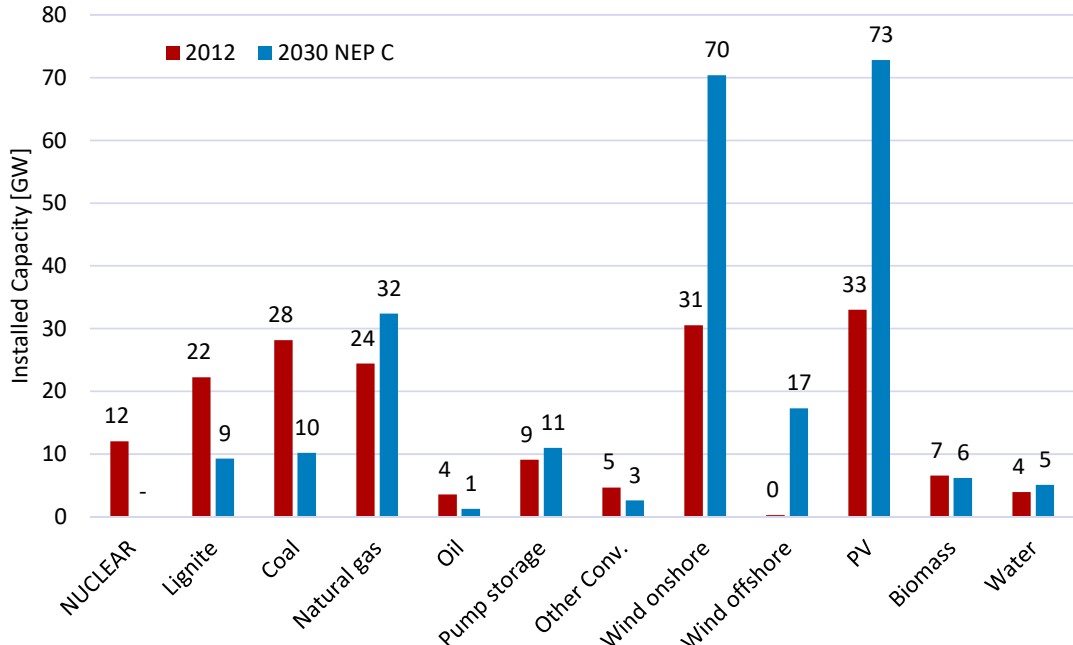

**Figure 4.** Aggregated power production capacities for Germany in 2012 vs. 2030 NEP C. Source: authors' assumptions based on [44].

The scenario devised includes the nuclear phase-out in Germany and the accelerated coal phase-out set to go into force, which entails the decommissioning of approx. 60% of the hard coal and lignite capacities operational in 2012. Furthermore, 8 GW of additional gas-fired power plant capacity is installed compared to 2012. In addition to these coventional capacities, the total installed capacities of wind and PV increase to 88 GW and 73 GW respectively. Similarly, we consider a rapid expansion of renewable energies in other European countries. In the proposed scenario, the price of $CO_2$ certificates increases to 29.4 €/ton. Overall, our scenario can be considered in line with the current objectives of the Paris agreement and national climate change mitigation objectives. The scenario describes a sustainable transition for the considered system and includes a high potential for an increase in the use of gas-fired power plants.

In contrast to fundamental changes in the electricity system, the composition of the gas sector exhibits more continuity. However, the German gas network also faces crucial challenges in the coming decade, especially regarding decreasing Dutch gas exports and new import pipelines such as Nord Stream 2. In order to enable a comparison of the 2012 and 2030 results, the decision is made to use the same historical time series of natural gas exports for 2030 as in 2012, while import capacities are adapted to 2030. We assume that the cross-border capacity from Russia to Germany via Nord Stream 1 and 2 amounts to 110 bcm/a, compared to 23 bcm/a in 2012. The second line of Nord Stream 1 was

finished in November 2012. Hence, in the simulation conducted for 2012 only 23 bcm/a of capacity is available. Additionally, we assume that German domestic production of natural gas is exhausted [45] and no further gas imports from the Netherlands in 2030 are possible as a halt in gas production at the Groningen field by 2030 has been announced [46]. Furthermore, we scale the of gas import price series in 2012 to an increased average gas price level of 26 EUR/MWh [44] in line with the scenario assumptions (cf. NEP 2030).

As described above, the two models, JMM and GAMAMOD-DE, are linked via the natural gas demand of German gas power plants. Consequently, all other sources of gas demand, i.e. household and industry demand, remain constant. It is worth noting that in this approach both the power market and the gas market are only considered as spot markets. This approach neglects that in the past European gas supply was dominated by long-term contracts. However, spot trading has gained in importance in European gas markets due to an increased share of LNG shipped gas quantities and an advanced degree European gas market integration [47]. We summarize the shared and model-specific assumptions of the two scenarios for 2012 and 2030 in Table 1.

**Table 1.** Data assumptions for 2012 and 2030.

| Category of Assumption | Parameter | Back-Testing 2012 | Future Scenario: Gas as a Bridge Fuel in 2030 |
|---|---|---|---|
| General assumption for both models | Temperature time series | 2012 | 2012 |
| JMM | German electricity imports and exports | Historical data (ENTSO-E) | Simulated |
| | Power plant capacities, Power transmission capacities | Historical data | EU: ENTSO-E TYNDP DE: NEP 2030 C |
| | Fuel & emission allowance prices | Historical data | NEP 2030 C |
| | Heat demand | Historical data | Historical data 2012 |
| | Time series of demand, renewable feed-in etc. | Historical data | Historical data 2012 |
| GAMAMOD-DE | German gas exports | IEA 2012 | IEA 2012 |
| | German gas imports | | No Dutch gas imports |
| | Gas import price | NCG natural gas quarter futures, 2012 | Scaled NCG natural gas quarter futures |
| | Infrastructure | Nord Stream I (23 bcm/a) | Nord Stream I + II (110 bcm/a) |
| | German gas production | | No German gas production |

Source: own assumptions, based on IEA [39]; ENTSO-E [6,43]; Datastream [40]; NEP [48].

## 3. Results and discussion

### 3.1. Backtesting 2012: Coupling JMM and GAMAMOD

The initial motivation for this study concerns the historical cold period of winter 2012, where shortfalls in the natural gas supply related to an elevated heat and electricity demand resulted in German gas power plants being shut down. Figure 1 compares the temperature time series of 2012 and the average residual load. During the first two weeks of February (1st to 13th of February), average temperatures were below −8 °C with the average residual load exceeding 68 GWh/h. As mentioned earlier, before simulating the year 2030 with similar time series, we evaluate our methodology and

calibrate the models by re-simulating the power and natural gas markets for 2012. In the following section, we compare the results of our backtesting with historical statistics.

The results of the German electricity market simulation for the year 2012 are presented in Figures 5–7. Figure 5 compares the adjusted statistical and simulated production mixes. As can be observed, the simulated power production by coal is slightly lower compared to historical values. At least part of the difference may be related to a different treatment of multi-fuel units and units with miscellaneous fuels such as blast furnace gas, which to our understanding is included in the coal-based generation in the IEA statistics.

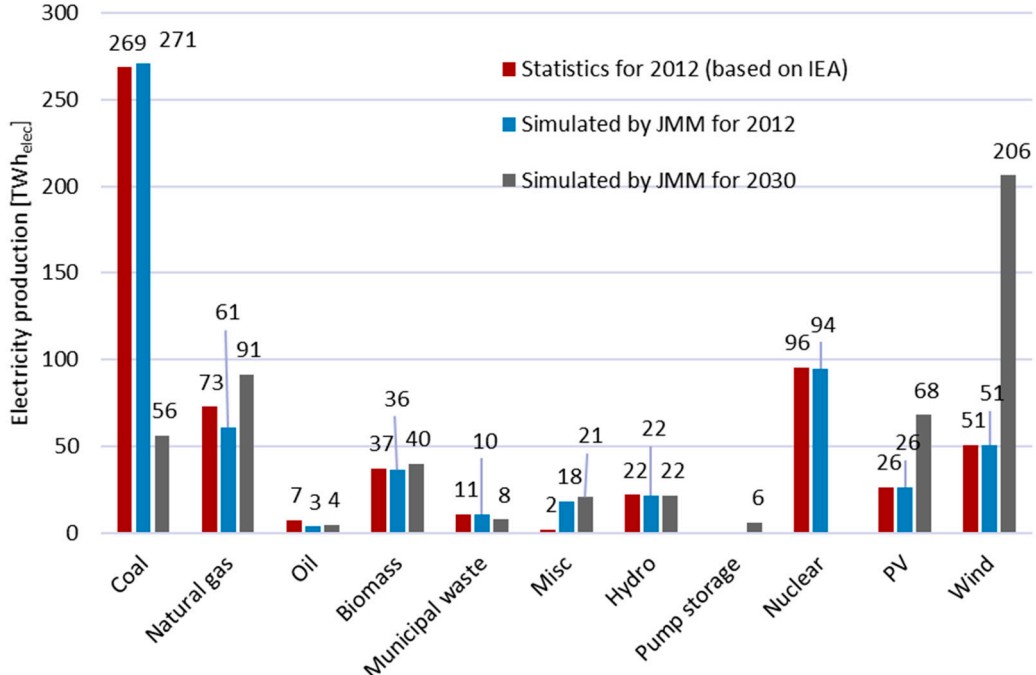

**Figure 5.** Results of JMM for the simulated annual production mix in the German power sector in 2012 (historic and simulated) and in 2030.

Additonally, the simulated natural gas power generation is 61 TWh$_{elec}$, which is 16% less than the historical value of 73 TWh$_{elec}$. This is considerably less than the typical deviation observed in electricity market models without consideration of CHP and the related heat generation. Yet the remaining discrepancy may be attributed to the following causes: First, the modeling is making use of random availabilities of power plants based on the average annual probability of technology outages and not the exact historical availabilities of each plant since these data are rather difficult to handle if available at all. Furthermore, we do not model redispatch which is increasingly used in Germany and which typically involves the activation of flexible natural gas power plants in the south of Germany and disadvantage coal power plants. Also, many gas-fired CHP plants are assumed to be connected to one aggregated district heating network, whereas in reality they are connected to various smaller networks. In periods with high heat demands, these gas power plants may be employed additionally. Also, we use the "planned" and not "physical" historical cross-border flows due to data availability restrictions. Lastly, there is a statistical difference in reported installed power plant capacity for 2012.

Figure 6 shows the natural gas demand in the power sector. The graph indicates that the highest electricity demand occurs on the 8 February 2012. This day has the highest historical daily-average price of electricity as can be seen in Figure 7. We also graph both the simulated and historical electricity prices for the winter period of 2012 in Figure 7. The development of both prices shows similar patterns, however, it is evident that the simulated prices do not reach the same peak as on 8 February. Achieving the exact historical price spike is very challenging by means of a fundamental model as a result of the

simplifying assumptions employed. The total simulated demand for natural gas in the power sector in 2012 is approx. 172 TWh$_{th}$.

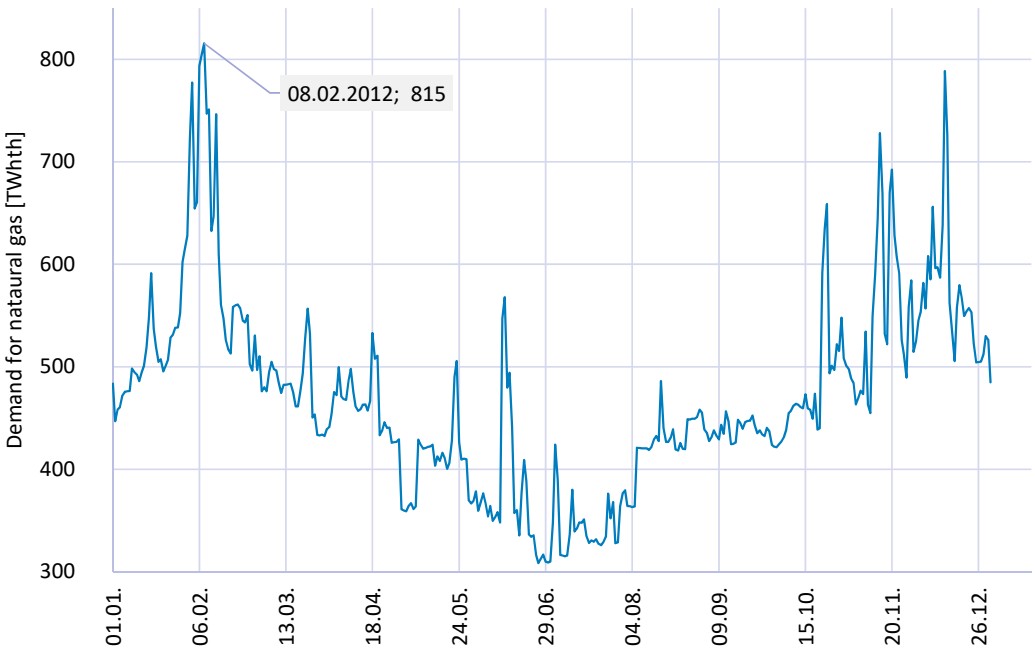

**Figure 6.** Results of JMM for the simulated daily consumption of natural gas in the German electricity sector for January-March 2012.

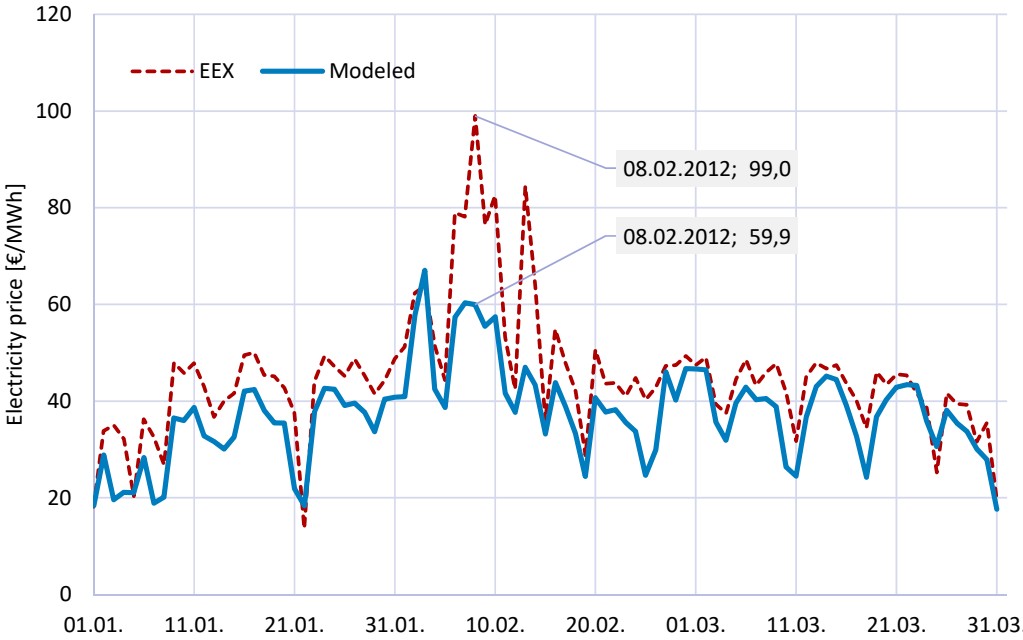

**Figure 7.** Average daily historical vs. simulated electricity prices for January-March 2012.

As mentioned earlier, we use the gas consumption of power plants calculated by JMM as input data for the model GAMAMOD-DE to resimulate the German gas market in 2012. Results of the re-simulation are illustrated in Figure 8, which shows the yearly gas balance in daily resolution. With the given demands and exports, GAMAMOD-DE optimizes the domestic gas production, import and storage operation using an objective function minimizing total system costs. Also, as the most expensive option, the model includes load shedding.

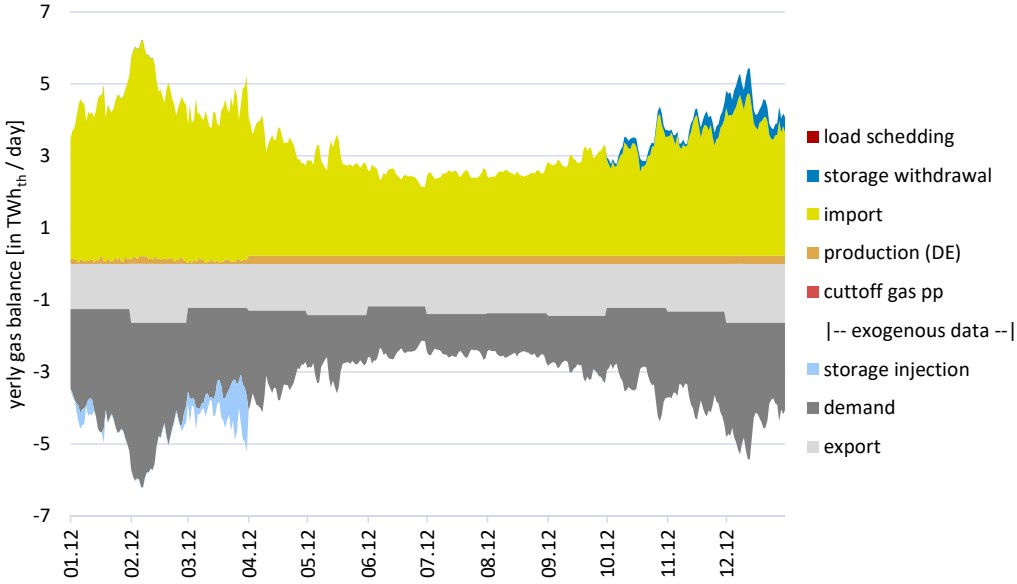

**Figure 8.** Results of GAMAMOD-DE for yearly natural gas balances in daily resolution in 2012.

As can be seen from Figure 8, German gas imports cover a significant part of demand and export during the year. While stable, domestic production only contributes a limited amount. Storage injection occurs in January and March while storages withdraw gas during the last three months of the year. Natural gas load shedding occurs, but as a percentage of the total gas balance it can be neglected (cf. below for detailed discussion).

Some effects in the modeled results deviate from general expectations of how gas markets should behave in real-world applications. First of all, gas storages generally use seasonal price differences in order to store natural gas during off-peak seasons, starting mainly in April, and withdraw gas during peak seasons, mostly during wintertime. In our approach, we took historical storage levels at the beginning of 2012 and assume the same storage levels at the end of 2012. As the model has the feature of perfect foresight regarding gas prices which increase from around 22 EUR/MWh$_{th}$ in January to 26 EUR /MWh$_{th}$ in December, the model tries to inject natural gas when gas prices are low and withdraw gas when prices are high or when supply cannot be covered by gas imports. The detailed monthly price data can be found in Appendix A.1. The model includes two restrictions on gas production, in particular, a daily cap and a yearly cap for each production location. Additionally, in accordance with [49], we introduced a flexibility parameter that allows higher daily production rates (+14%) in order to provide higher flexibility. However, the yearly production cap is still a binding condition. Hence, the model saves gas production quantities at the beginning of the year when gas prices are low and uses the maximum flexibility of German production facilities at the end of the year when prices for gas imports are at their highest level. Due to low domestic gas reserves, Germany depends on natural gas imports [50]. In 2012, ca. 27% of gas supplies were sourced from the Netherlands (NL) and 26% from Norway (NO). The most important supplier is Russia (RU), supplying 45% of natural gas via pipelines through the Baltic Sea (Nord Stream I), and via transit countries, i.e. Belarus, Poland (PL), Ukraine, Slovakia, Czech Republic (CZ) and Austria (AT). A comparison of historical 2012 gas imports and model results can be found in Appendix A.2.

The total modeled volume of load shedding in 2012 amounts to 90 GWh/a with peaks during February and November when average temperatures fell below zero degrees centigrade. As can be seen in Figure 9 regarding non-supplied gas, the model cuts 77 GWh /a or 0.04% of the total gas demand of the power plants. We assume the lowest cost for load shedding of gas power plants (90 EUR/MWh), as some of them have interruptible contracts. Following [49], we assume the cost for gas supply interruptions in industries to be 93 EUR/MWh (up to a volume of 50% of the demand) and

180 EUR/MWh for private households. Hence, the share of industry demand curtailed by the model in 2012 is limited to (13 GWh/a) and for households to less than 1 GWh/a.

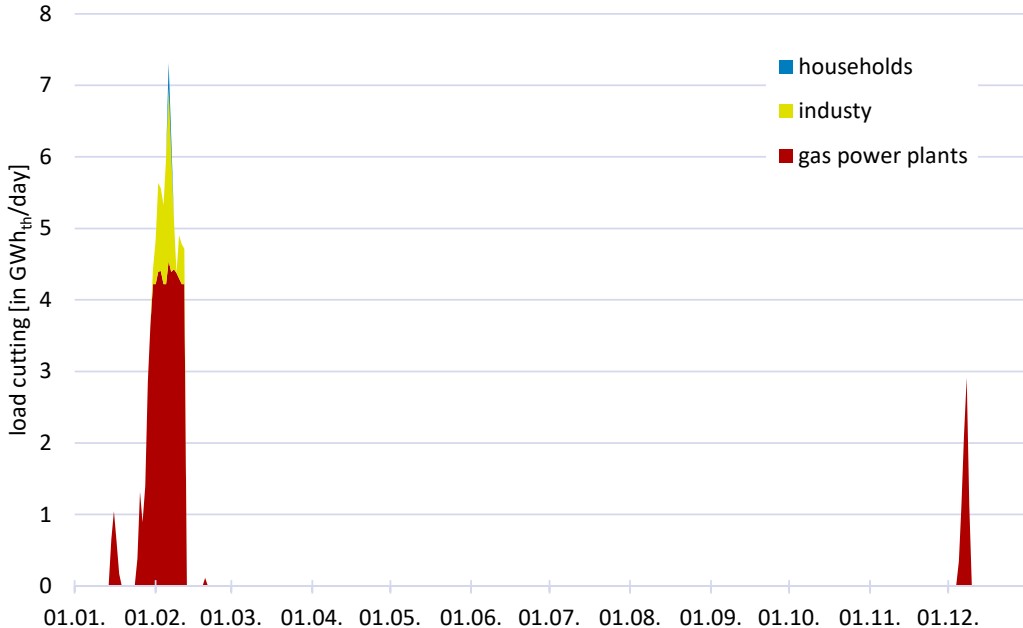

**Figure 9.** Results of GAMAMOD-DE for load shedding in Germany in daily resolution for 2012 aggregated by sector.

On the 6th of February 2012, the coldest day in 2012, the model results indicate load shedding of 7.3 GWh$_{th}$. Figure 10 illustrates the high utililization rate of pipelines in the northeastern part of Germany, where some pipeline capacities are fully used. Some smaller pipelines in the south of Germany transport gas with capacity utilization rates between 50–100% of their initial transport capacity. However, the incidence of fully utilized pipelines leads to shortfalls only in the northeastern part of the German gas network, namely in the grid area of the transmission system operator ONTRAS. However, it should be noted that these results depend on our model assumptions and although historic shortfalls are reported that led to load shedding of 3.5 GW of natural gas power plant demand during February 2012, the affected grid areas, Open Grid Europe, Thyssengas and Energienetze Bayern, are located in the south of Germany [51].

Coupling the models based on the year 2012, we observe slight differences in the generation of gas power plants (in JMM) and in gas import balances (in GAMAMOD-DE). These differences can generally be explained by the lack of data, necessary model simplifications, e.g., limitation of the geographical scope. Nevertheless, the approach established that the integration of the natural gas market and electricity market models provide a sufficient basis for simulating the system in 2030 and to offer insights into the implications of a stronger coupling of the gas and electricity system for energy security in extreme weather situations.

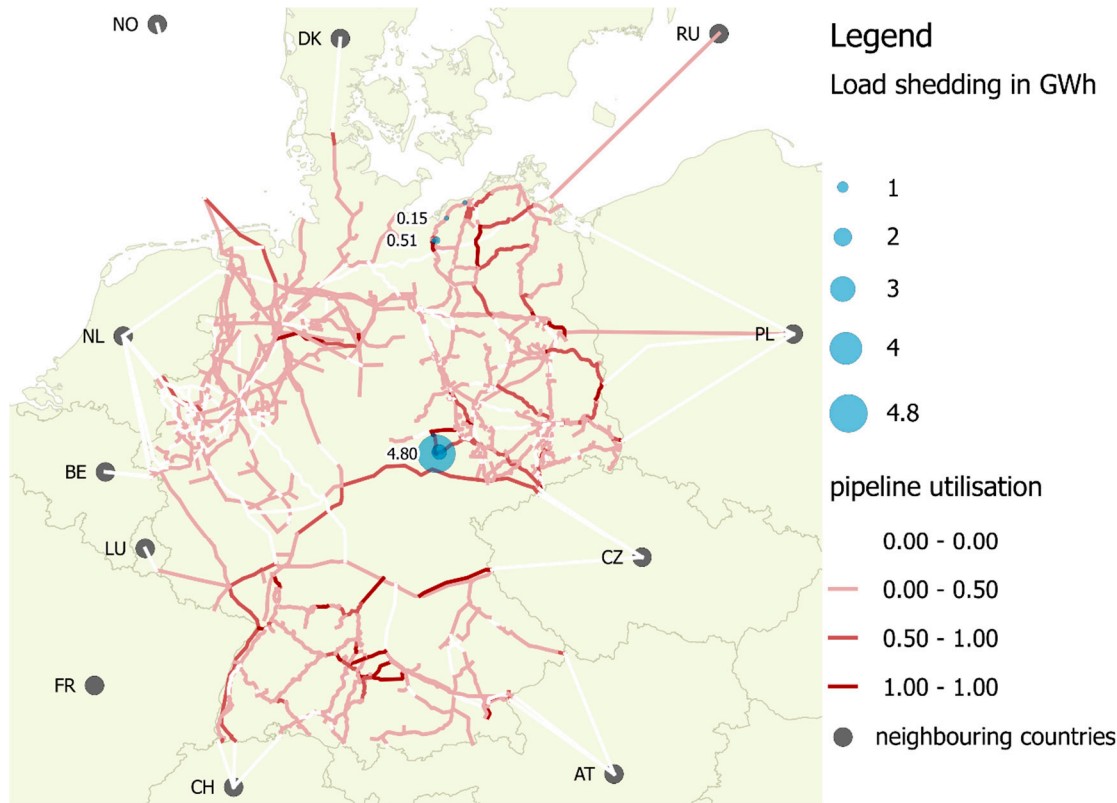

**Figure 10.** Results of GAMAMOD-DE on simulated German load shedding regions in 2012.

### 3.2. Increased Installed Capacity of Gas Power Plants Drives Gas Demand in 2030

Figure 5 illustrates not only the simulated generation mix in 2012, but also for the German electricity market in 2030. As a consequence of the considered assumptions, electricity production from intermittent renewables, namely wind and PV, notably increases from 77 TWh in 2012 to 274 TWh in 2030. Also, electricity production from coal decreases by 80% due to the high feed-in of renewables and the accelerated decommissioning of aged plants. In this scenario, renewables cover a great part of the load and gas power plants act as the central back-up technology. Consequently, gas-fired plants produce 91 TWh$_{elec}$ of electricity, which is 50% higher than in 2012. Figure 11 highlights the development of daily demand for natural gas in the electricity sector. We observe a high natural gas demand in the period between 27 January and 15 February. The average daily demand in this period is 1237 GWh in 2030, which is 80% higher than the average demand for the same period in 2012 (approx. 670 GWh). Furthermore, the peak demand is approx. 1400 GWh on 8 February 2030, which is 71% higher than the peak demand on 8 February 2012. The total demand for natural gas in the power sector is approx. 260 TWh$_{th}$ in 2030, which is 51% higher than its value in 2012.

### 3.3. German Gas Network with Limited Increased Load Shedding Exhibits Resiliency in 2030

As outlined above, the results of the electricity market simulation in 2030 show an increased natural gas demand for gas power plants. As we assume no change in the gas demand for other sectors (private households and industry) and exports compared to 2012, the total gas demand increases slightly in 2030 due to the increase in power sector demand. The aggregated gas balances for 2012 and 2030 are shown in Figure 12. Both balances differ only slightly. Firstly, we assumed no further domestic gas production in Germany in 2030. The increased gas demand in the power sector (+93 TWh) and the missing domestic gas production (−74 TWh) are compensated by higher gas imports (+166 TWh) compared to 2012. While in 2030, according to our assumptions Dutch imports cease, Norwegian and Russian gas imports increase their share in the supply of the German gas market. Although the

absolute volume of load shedding is small (907 GWh/a), the amount is still more than ten times higher than in 2012.

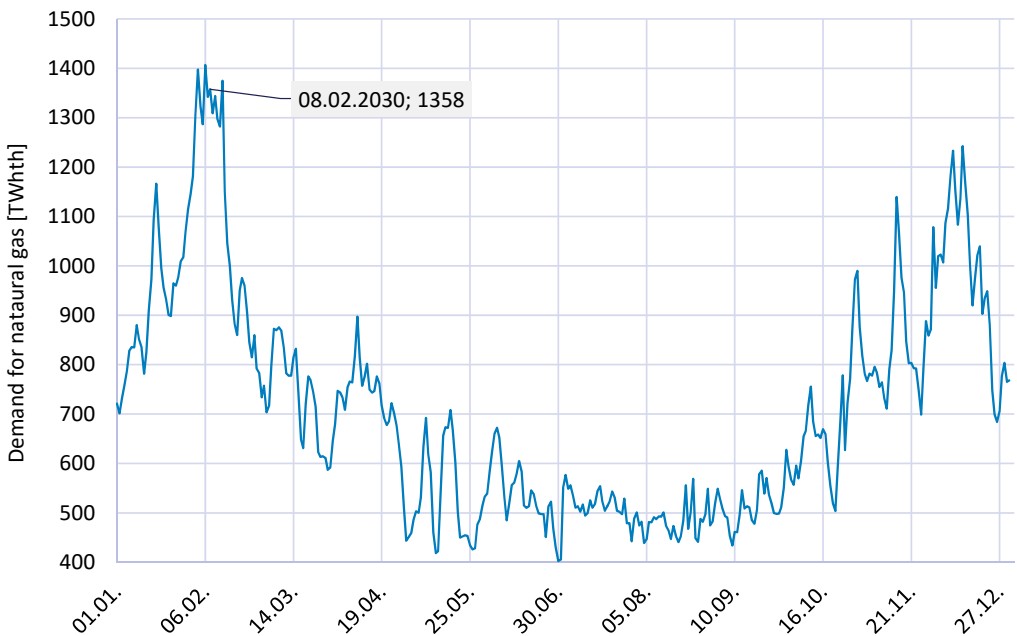

**Figure 11.** Results of JMM for simulated daily demand for natural gas in the German power sector in 2030.

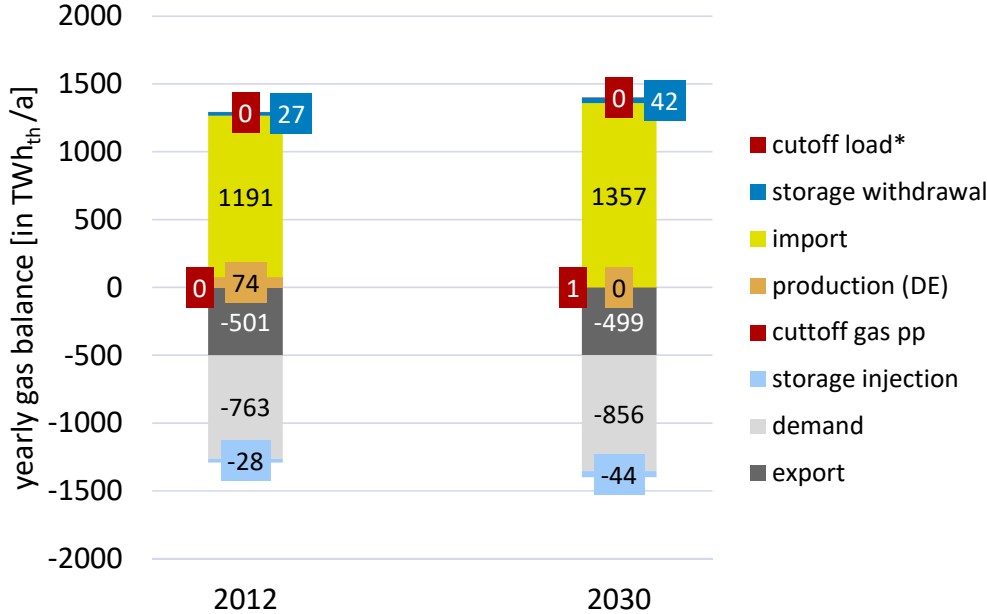

**Figure 12.** Results of GAMAMOD-DE for yearly gas balances in 2012 and 2030.

Figure 13 shows the total aggregated load shedding clustered by demand sectors. The highest levels of load shedding occur for gas power plants and small CHP plants, but also the share comprising industry and private households increases. As load shedding of gas power plants represents the least expensive option in the model, it is deployed most often in the cost-minimization problem. However, load shedding does not occur permanently throughout the year, but only on days with high gas demands. Figure 14 illustrates daily load shedding in 2030 differentiated by demand sector. With the assumed temperature curve of 2012 in mind, load shedding of more than 10 GWh$_{th}$ per day occurs especially during wintertime, at the beginning of February and December when temperatures

are below zero degrees (cf. Figure 1). Thus, load shedding of small CHPs in 2030 could potentially endanger energy security. They not only provide electricity but also supply heat at a local level. While a shortfall in the gas supply for gas power plants might be compensated by other power plants in the transmission grid, a disruption in local heat production cannot be compensated easily (cf. discussion in Section 3.4)

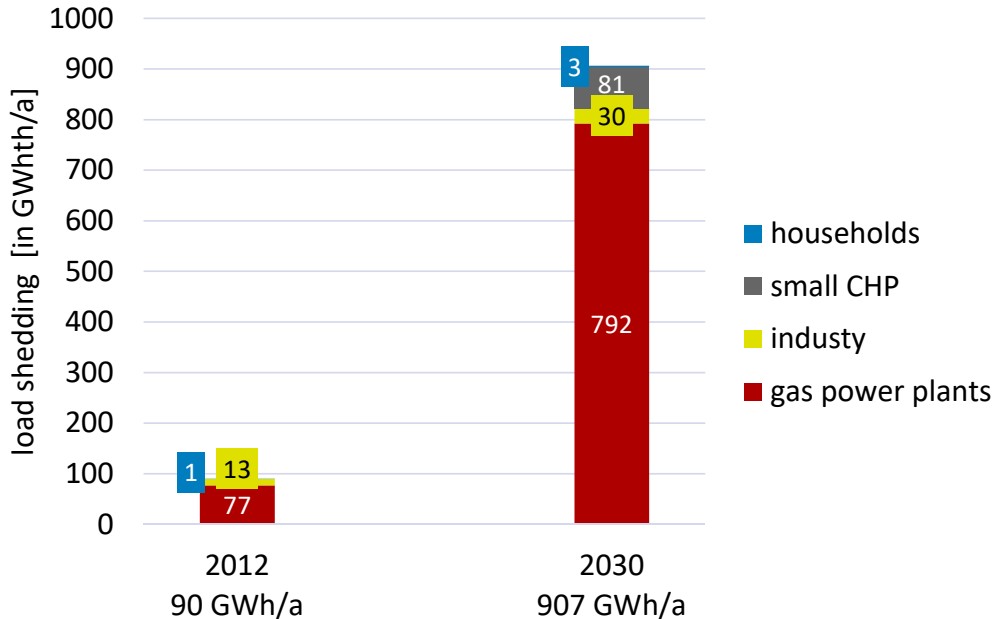

**Figure 13.** Results of GAMAMOD-DE for load cutoff in 2012 and 2030, clustered by gas demand sectors.

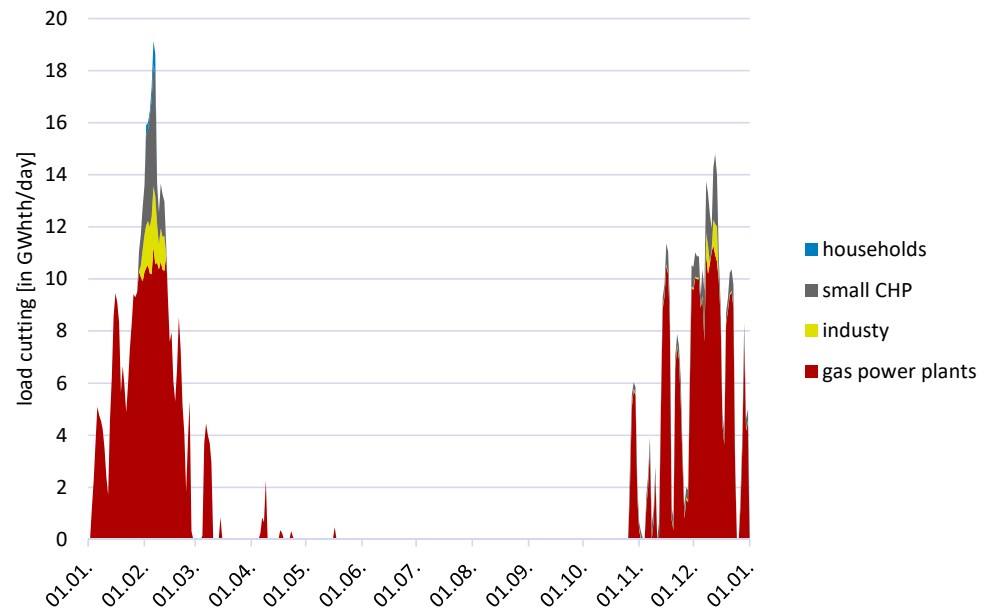

**Figure 14.** Results of GAMAMOD-DE for load shedding in 2030, clustered by gas demand sectors on a daily resolution.

In GAMAMOD-DE, due to the assumed cost structure, load shedding represents the last resort to ensure model feasibility. In real-world applications, grid operators would resort to these options to ensure the operational integrity of the system in order to meet the demand in their network area. However, high utilization rates may also be critical for energy security, as a system operating at full capacity cannot react to unanticipated events, e.g., unplanned technical outages, environmental

disasters, etc. Therefore, an analysis of pipeline utilization can provide an indication of gas grid reserve capacity. Figures 15 and 16 show pipeline utilization on the 6th of February with low daily average temperatures (−13 °C) and high shares of load shedding (19 GWh$_{th}$,). In general, a smaller number of pipelines is completely (100%) or strongly (50–100%) utilized.

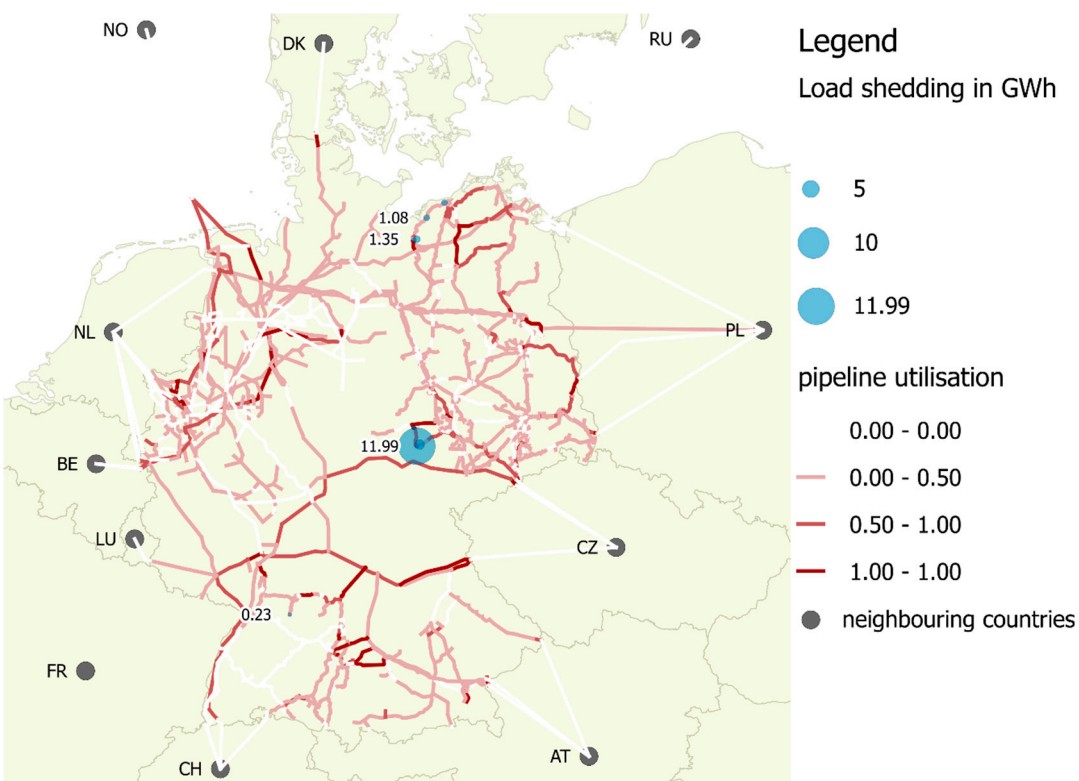

**Figure 15.** Results of GAMAMOD-DE for pipeline utilization on the 6th of February 2030.

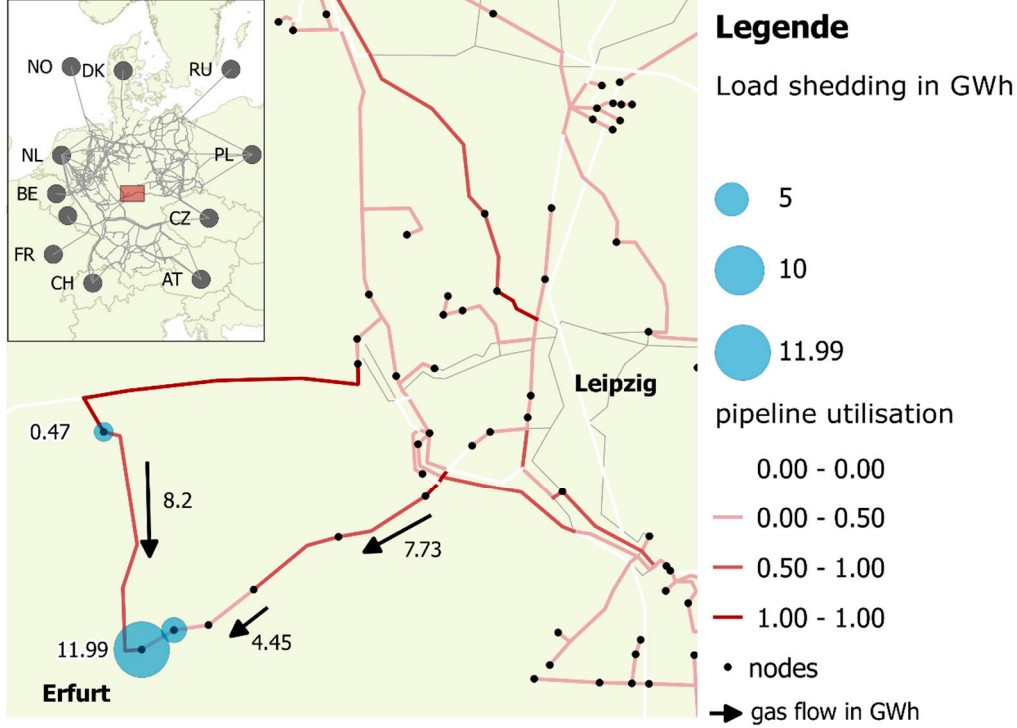

**Figure 16.** Results of GAMAMOD-DE for pipeline utilization on 13th of December 2030.

At the demand peak on the 6th of February, gas pipelines close to neighboring eastern European countries, Poland and the Czech Republic, are fully utilized. Load shedding occurs at the same locations as in 2012, but to a larger extent. The highest level of load shedding occurs at an exit node close to Erfurt, where almost 12 GWh of load shedding occurs, consisting of 10.6 GWh for gas power plants and 0.4 GWh for smaller CHPs, 0.9 GWh for households and 0.04 GWh for industries.

Figure 16 provides a detailed depiction of load shedding in the grid area close to Erfurt. The infrastructure shows a coarsely meshed gas grid that leads to congestion in the assumed situation with very cold temperatures and an additional gas demand arising from newly installed small CHP plants.

A general analysis of pipeline utilization in 2030 provides two insights. Firstly, a number of gas pipelines exhibit a low utilization rate throughout the year. One reason might be that they provide n-1 security for the system. However, if gas usage fluctuation increases, the issue of disinvestment could arise, especially in regions where gas can be substituted easily. Secondly, there are a small number of pipelines with high utilization rates (> 50%) throughout the entire year. An analysis of the SoS should carefully consider these particular pipelines as a disruption could create larger shortfalls in the entire gas system. Therefore, further analyses should focus on unanticipated critical situations in the gas network (e.g., unplanned outages of pipelines, unexpected cold weather events) to assess the resiliency of the energy system in these situations.

### 3.4. Implications of Gas Grid Congestion for the Electricity System

Our results have shown that isolated extreme weather events can occur where the gas demand of gas power plants cannot be met in a timely manner. A detailed investigation of the effects of natural gas congestion on the electricity market requires additional iterations between the models which can be performed in future studies. However, in this section, we discuss some effects of gas supply interruption on the security of power and heat supply.

In 2012, load shedding in the natural gas network interrupted the gas supply to two gas nodes. The first node is located in the northern part of Germany with one connected power plant with a capacity of 23 MW. The second node is in the federal state of Thuringia (central part of Germany) with two connected power plants with capacities of 11 and 76 MW, respectively. The disrupted electricity generation could easily be substituted with other gas plants in different locations or with more expensive units such as oil-fired plants, etc. The two power plants in the Thuringia region are CHP units and, to our knowledge, the only CHP plants that provide heat to the Erfurt heating network. In the case of an interruption of the gas supply, back-up boilers could generate heat, however, these boilers are mainly gas-fired. Accordingly, the security of heat supply could potentially be endangered in the case of load shedding in the natural gas network. The gas power plant in the northern part of Germany is also a CHP unit. As this unit is part of one of the aggregated heating networks, an increased utilization of other units within the heating network can compensate for the disrupted heat generation in the modelling exercise—although this is hardly feasible in reality.

In 2030, load shedding in the natural gas network interrupted the gas supply to the same gas nodes. The difference is that in this year new gas power plants are installed and allocated to the Thuringia heating network. Consequently, the magnitude of load shedding is higher. In 2030, similar to 2012, other plants with the same operational costs compensate for the interrupted electricity generation. The incidence of gas load shedding, however, still poses a threat to the security of heat generation.

## 4. Limitations of this Study

Assumptions and simplifications that pose limitations to our study are discussed below. We use a two-stage approach, starting with the power market and subsequently utilizing a part of the results as input data for the gas market model. This sequential nature of the modeling approach does not consider the equilibrium between both sectors. Hence, our study neglects feedback effects of possible gas congestion or load shedding in the natural gas networks as regards the dispatch of gas power

plants and thus, the actual gas demand of gas-fired power plants. One possible solution might be an iterative approach (cf. [30]).

The scenario employed does not include additional flexibilities in the electricity system, e.g., electric vehicles or power-to-heat and power-to-gas technologies. Accordingly, natural gas power plants are operating additionally to provide flexibility in the electricity sector and balance the demand and supply from intermittent renewables. Considering further flexibility options might reduce the operation of gas-fired power plants and ease possible congestion in the gas network. In the heating sector, we consider only a select number of big district heating networks (approx. 28 district heating networks) and aggregate the rest of the heating networks into a few large networks. Also, we do not consider behavioral effects.

For GAMAMOD-DE, three major limitations should be mentioned. Firstly, the covered period in the model assumes equal storage levels at the beginning and end of the modeled period of one year (January to December). As storages are almost full in January, the model tends to inject natural gas into the storages at the end of the model period, from November to December. This kind of storage operation differs from real-world observations. One possible solution might be to change the modeled horizon from April to April, where storage levels start and end at a lower level. Secondly, GAMAMOD-DE considers perfect foresight during the optimization horizon. Hence, it already anticipates cold periods in the beginning. This is crucial for facilities that provide back-up capacities for the gas system, e.g., gas storages that provide sufficient capacity during cold days. An alternative approach would be to run the optimization two times while fixing storage operation results of the first run based on non-critical temperatures. For the second run, the temperature time series of the cold weather event would be used. Consequently, the fixed injection and withdrawal rates to and from storages would be shocked by higher demand and the flexibility of the pipeline system could be evaluated in a non-anticipative environment (cf. [14,15]). Finally, the model simplifies gas flows and neglects the physics of gas transportation, e.g., line packing that could be covered by modeling the Weymouth equation. However, this would lead to non-linear constraints, complicating the solution of an optimization model.

Our study indicates the need for further research on the role of natural gas in the context of the Energiewende. While Gillesssen et al. argue that energy security will increase in the horizon of 2050 due to a decreasing natural gas demand [31], we find that natural gas in the power sector might increase in the mid-term until 2030 and will then challenge the existing gas network infrastructure. Technical constraints seem not to preclude additional gas demand in the power sector, thus, the question arises as to whether investment and construction of new gas power plants will take place in a timely manner. Furthermore, fluctuating gas demand may lead to the underutilization of gas grids and increase economic pressure on grid tariffs. Hence, an evaluation of investments in gas grid infrastructure in the context of potential stranded assets should be considered an essential research topic in the future.

## 5. Conclusions

In the preceding analysis, we apply a novel approach by coupling the unit commitment electricity market model JMM with a gas market model representing the Germany natural gas network, GAMAMOD-DE. We initially evaluate our method by re-simulating the extreme weather event in 2012 (cold Dunkelflaute). Furthermore, we use the approach to simulate the combined electricity and natural gas sectors in a scenario reflecting an advanced stage of the energy transition in 2030 to study the security of gas supply by identifying possible incidences of network congestion. The result of the power system simulations suggests a 51% increase in the yearly natural gas demand in the power sector in 2030 compared to 2012. We also observe that on an extremely cold winter day (8 February), the demand for natural gas in the power sector increases by 71% in 2030 compared to 2012. As pertains to the natural gas sector, the results of the study suggest that in 2012 the shortfalls of supply to gas power plants were not caused by technical network congestion, but other drivers, e.g., political or economic reasons. Regarding an enforced sector coupling in 2030, only slightly increased load shedding occurs, but still at a relatively low level. Regional hot spots of these incidences of load shedding are located in

the federal state of Thuringia, where pipeline meshing is coarse and higher additional demand from gas power plants is expected. Consequently, extensions of individual pipelines may mitigate shortfalls in this area. However, current projects of common interest (PCI) in Germany include activities only for the Trans-Europa-Naturgase-Pipeline (TENP) located in Baden-Wuerttemberg [52]. Regarding the implications of possible shortfalls in the gas sector for the electricity sector, we find that other gas power plant capacities are available to balance electricity supply. There is, however, a lack of local heat production from small CHPs, which might be affected by load shedding, potentially posing a threat to a secure heat supply. In conclusion, this study indicates that the natural gas system in Germany appears to be robust against an increased demand from natural gas power plants. However, results for the Erfurt region suggest the need for further studies on the threat of heat supply disruptions.

**Author Contributions:** P.H. and S.H. led the coding and modeling efforts as well as data research and writing the paper. Additionally, P.H. managed the reviewing and editing process. C.W. and D.M. contributed to the model implementation as well as supervision, proof-reading, discussing results and implications and the editing process. Furthermore, they initiated the research, supported policy backgrounds, and contributed to writing the text.

**Funding:** This research was funded by Federal Ministry of Economic Affairs and Energy, through the project "LKD-EU", grant number 03ET4028C and 03ET4028D.

**Acknowledgments:** We thank all members of the LKD-EU project team: Friedrich Kunz, Alexander Weber, Mario Kendziorski, Wolf-Peter Schill, Christian von Hirschhausen, David Schönheit, and Jakob Mucke for discussion and fruitful comments. We also thank the participants of the final LKD-EU workshop as well as Matthew Schmidt and Ruud Egging. We acknowledge support by the Open Access Publication Funds of Technische Universität Dresden.

**Conflicts of Interest:** The authors declare no conflict of interest. The funders had no role in the design of the study; in the collection, analyses, or interpretation of data; in the writing of the manuscript, or in the decision to publish the results.

## Appendix A

*Appendix A.1 Cross-Border Natural Gas Prices in 2012 and 2030*

In GAMAMOD-DE, we assume uniform cross-border natural gas prices for gas imports in accordance with the quarter futures price in the gas market area Net Connect Germany (NCG) in 2012 [40]. The average NCG natural gas futures gas price amounts to 24.56 EUR/MWh$_{th}$. For 2030, we scale the annual price pattern of 2012 up to the average gas price of 26 EUR/MWh$_{th}$, based on [44] in order to be consistent with the assumptions made in JMM. Table A1 shows the assumed values for 2012 and 2030 of German cross-border quarter natural gas futures prices.

**Table A1.** Cross-border natural gas prices based on NCG natural gas quarter futures for 2012 and own assumptions for 2030.

| Month | Price 2012 [in EUR/GWh$_{th}$] | Price 2030 [in EUR/GWh$_{th}$] |
|---|---|---|
| January | 22,400.00 | 24,067.97 |
| February | 22,400.00 | 24,067.97 |
| March | 22,400.00 | 24,067.97 |
| April | 25,350.00 | 27,247.82 |
| May | 25,350.00 | 27,247.82 |
| June | 25,350.00 | 27,247.82 |
| July | 24,050.00 | 25,850.50 |
| August | 24,050.00 | 25,850.50 |
| September | 24,050.00 | 25,850.50 |
| October | 26,430.00 | 28,408.68 |
| November | 26,430.00 | 28,408.68 |
| December | 26,430.00 | 28,408.68 |

*Appendix A.2 Comparison of Simulated and Historical Gas Imports in 2012*

Figure A1 shows the modeled and historical import quantities, based on [50]. The model overestimates gas imports from Russia via the Czech Republic and underestimates Norwegian gas imports. One reason might be long-term contracts that are not included in the model.

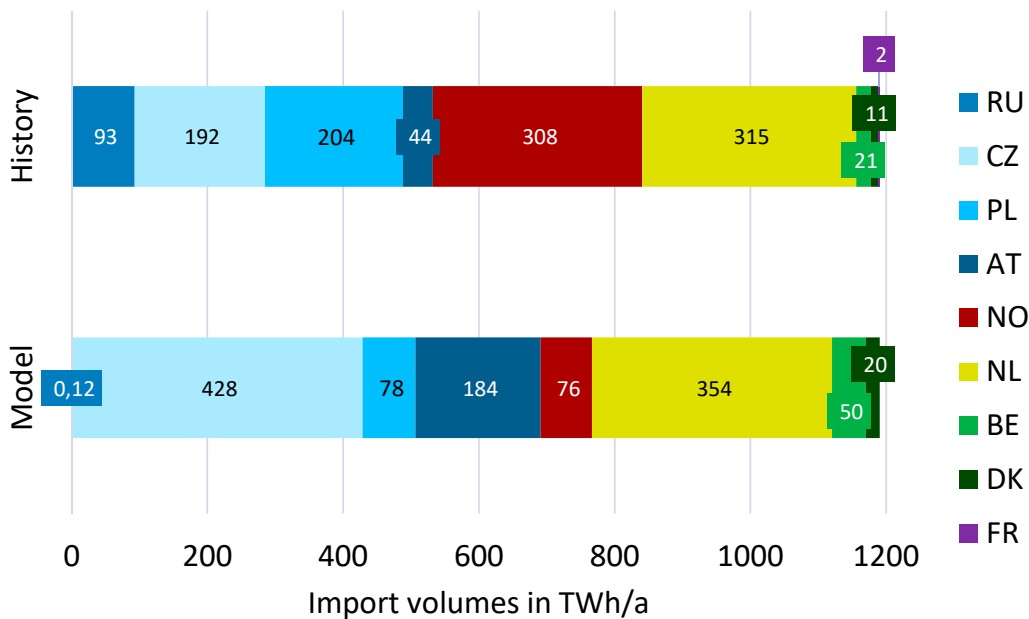

**Figure A1.** Historical and simulated German gas imports in 2012.

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
