# Peer review of "Does Increasing Natural Gas Demand in the Power Sector Pose a Threat of Congestion to the German Gas Grid? A Model-Coupling Approach"

_energies, doi:10.3390/en12112159_

Round 1

Reviewer 1 Report

This is an interesting study. The methdology and data used in this study were well described. The interpreation of results is also clearly presented. The only problem is that there are several mistakes in English expression and grammar. I encourage authors to receive English corrections with help of native English writer. After these minor revisions, this manuscript can be published. 

Author Response

Please, find the answer in the attached file.

Reviewer 2 Report

General comment:This paper deals with the modelling of the electricity system in Germany.   

The Abstract is not well written. You should rewrite it focusing on the aim of the paper, main methods, main results and few recommendations based on empirical results. 

Introduction:The authors should improve the part of introduction by describing more the issues from literature.  

Methodology: Present in the methodological part details regarding the advantages and limits of the methods used in this research. Alternative methods should be specified. 

Results:The empirical analysis is quite superficial. Maybe forecasts for more years and comparisons between future evolutions of the indicators could be more suggestive. Comparisons with results from previous studies are necessary. 

Discussion:Interpretations of the results are provided, but a more critical position is required. Provide more comments on data and results.  

Bibliography/References:Extend the references with recent ones. 

Other remarks:

Decision: Accept with major corrections.

Author Response

Please find the answer in the attached file.

Round 2

Reviewer 2 Report

Accept in present form